# *Mycobacterium tuberculosis* impairs protective cytokine production via transcription factor MafB manipulation

Hiroyuki Saiga[1,2,3]*, Masaki Ueno[4], Toshiki Tamura[3], Yusuke Tsujimura[3], Masamitsu N. Asaka[3], Yumiko Tsukamoto[3], Tetsu Mukai[3], Michito Hamada[5], Satoru Takahashi[5], Takashi Tanaka[6], Tsuneyasu Kaisho[7,8], Yoshimasa Takahashi[2], Katsuaki Hoshino[1,8], Manabu Ato[3]

**1** Department of Immunology, Faculty of Medicine, Kagawa University, Kagawa, Japan, **2** Research Center for Vaccine Development, National Institute of Infectious Diseases, Japan Institute for Health Security, Tokyo, Japan, **3** Leprosy Research Center, National Institute of Infectious Diseases, Japan Institute for Health Security, Tokyo, Japan, **4** Department of Pathology and Host Defense, Faculty of Medicine, Kagawa University, Kagawa, Japan, **5** Department of Anatomy and Embryology, Institute of Medicine, University of Tsukuba, Ibaraki, Japan, **6** Laboratory for Developmental Genetics, RIKEN Center for Integrative Medical Sciences (IMS), Kanagawa, Japan, **7** Industry-Government-Academia Collaboration Promotion Headquarters, Wakayama Medical University, Wakayama, Japan, **8** Laboratory for Human Disease Models, RIKEN Center for Integrative Medical Sciences (IMS), Kanagawa, Japan

* saiga-h@niid.go.jp

## Abstract

Although an increased expression of the transcription factor v-maf avian musculoaponeurotic fibrosarcoma oncogene homolog B (MAFB) has been reported in patients with active tuberculosis (TB), its potential role in *Mycobacterium tuberculosis* infection remains unknown. Herein, we report that MafB in macrophages is a regulator of the pro-inflammatory cytokines, TNF-α and IL-12p40, which are crucial for host defense against *M. tuberculosis* infection. Cell-based luciferase assays showed that MafB inhibited TNF-α and IL-12p40 transcriptional activity in a dose-dependent manner. At the molecular level, MafB interacted with IFN regulatory factor (IRF)-5 and PU.1 and inhibited IRF-5- and PU.1-mediated transactivation, via the basic-leucine zipper domain. Analysis using gene-deficient macrophages demonstrated that the suppressed pro-inflammatory cytokine production during *M. tuberculosis* infection depends on MafB expression. Finally, *in vivo* studies indicated that *M. tuberculosis*-mediated increase of MafB expression was responsible for the exacerbation of *M. tuberculosis* infection. Thus, our results provide a functional view of MafB as a cytokine regulator as well as novel insights into host factors involved in TB susceptibility.

## Author summary

The WHO Global tuberculosis report for 2024 states that tuberculosis has replaced COVID-19 as the leading cause of death from a single infectious agent

**Data availability statement:** All relevant data are within the manuscript and its Supporting Information files.

**Funding:** This work was supported by JSPS KAKENHI Grant Number JP17K15665 to H.S., JP19K07482 to H.S., JP22H04922 (AdAMS) to H.S., and JP23K07933 to H.S. and by AMED under Grant Number JP24fk 0108648 to Y. Tsujimura and JP25fk0108673 to M.A.. This work was also supported by grants from the Takeda Science Foundation (to H.S.) and the Next Generation Leading Research Fund for 2017 of the Kagawa University Research Promotion Program (to H.S.). The funders had no role in study design, data collection and analysis, decision to publish, or preparation of the manuscript.

**Competing interests:** The authors have declared that no competing interests exist.

globally. Therefore, there is a need to identify the hallmarks of active tuberculosis. An increased expression of the transcription factor MAFB has been reported in patients with active tuberculosis; however, the significance of MafB in *Mycobacterium tuberculosis* infection remains poorly understood. Here, we demonstrated the critical role of MafB in macrophages in *M. tuberculosis* infection. We observed a negative correlation between the expression of MafB and induction of pro-inflammatory cytokines, which are essential for host defense against *M. tuberculosis* infection. We also demonstrated that elevated MafB expression after *M. tuberculosis* infection reduced pro-inflammatory cytokine production and anti-tuberculosis activities, promoting mycobacterial survival and persistence. Thus, *M. tuberculosis* creates a hospitable environment for itself by regulating the expression of MafB. Our findings provide insights into a unique strategy for mycobacterial survival and the pivotal role of MafB as a biomarker for tuberculosis susceptibility.

## Introduction

Tuberculosis (TB) is a re-emerging infectious disease caused by the acid-fast bacillus *Mycobacterium tuberculosis*. One-fourth of the global population is estimated to be infected with *M. tuberculosis*; most are asymptomatic (termed latent TB). However, 12 million people developed active TB, resulting in 1.25 million deaths in 2023 [1,2]. The recent increase in the number of TB cases, partly owing to COVID-related disruption of TB care, has led to an urgent need to identify novel hallmarks of this threatening disease.

Genome-wide association studies (GWASs) provide information regarding susceptibility genes involved in the onset of disease via a comprehensive analysis [3]. Several studies have identified loci likely contributing to TB susceptibility in East Asia and identified the transcription factor v-maf avian musculoaponeurotic fibrosarcoma oncogene homolog B (MAFB) as a candidate gene for TB susceptibility [4–6]. In addition, genome-wide transcriptional profiles generated from patients with active and latent TB, as well as healthy controls, in the UK and South Africa defined a distinct 393-transcript signature in patients with active TB. Notably, MAFB was included in this 393-transcript list [7]. Thus, MAFB is a common candidate gene for TB susceptibility across racial groups; however, its functional consequences for TB remain poorly defined.

MafB, a member of the large Maf transcription factor family, which is essential for tissue development and cell differentiation, is expressed in various tissues and cell types and regulates distinct target genes depending on the cell type [8–11]. MafB is expressed in tissue macrophages and plasmacytoid dendritic cells (pDCs) [12–14]. Additionally, an integrative genomic analysis of gene expression estimates that MafB may be broadly involved in intracellular anti-inflammatory activities [15]. Among them, several studies on the regulation of type I interferon by MafB mention its role in autoimmune diseases such as psoriasis and viral persistence in patients with chronic

hepatitis C [13,16,17]. Furthermore, gene-targeting experiments demonstrated that MafB directly regulates the complement component C1q, leading to the prevention of systemic lupus erythematosus-like autoimmune diseases [18]. These findings suggest that MafB broadly functions as a regulator in the immune system and could contribute to the pathogenesis of various autoimmune and infectious diseases.

In this study, we investigated the role of MafB in macrophages. Our data demonstrate its role in regulation of pro-inflammatory cytokine production. In addition, we found MafB-dependent restriction of inherent macrophage activities against *M. tuberculosis* infection using macrophage-specific *Mafb*-deficient mice. Our findings not only elucidate novel molecular subjects involved in pro-inflammatory cytokine regulation but also provide insights into a unique strategy for mycobacterial survival and in host factors which determine TB susceptibility.

## Results

### MafB suppresses the transcriptional activation of TNF-α and IL-12p40

A chromatin immunoprecipitation-sequencing (ChIP-seq) analysis of the transcription factors in bacterial lipopolysaccharide (LPS)-stimulated macrophages revealed that NF-κB p65, IFN regulatory factor (IRF)-5, and PU.1 strongly contributed to the Toll-like receptor (TLR)-mediated induction of pro-inflammatory cytokines, including tumor necrosis factor (TNF)-α and interleukin (IL)-12 [19]. We first constructed expression plasmids containing these transcription factors and performed a luciferase assay using heterologous HEK293T cells to examine the role of MafB in the transactivation of the pro-inflammatory cytokine genes (Fig 1A and 1B). Although TNF-α transcriptional activity driven by either IRF-5 or PU.1 was low, co-expression of NF-κB p65, IRF-5, and PU.1 synergistically activated *Tnf* promoter (Fig 1A). Similar results were obtained when cells were transfected with the *Il12b* luciferase reporter (Fig 1B), demonstrating that the transcription factors, NF-κB p65, IRF-5, and PU.1 are important for the induction of TNF-α and IL-12p40. Under these conditions, we found that MafB reduced TNF-α and IL-12p40 transcriptional activity in a dose-dependent manner (Fig 1A and 1B). This result suggests that MafB suppresses the activation of TNF-α and IL-12p40. Subsequently, we examined TNF-α and IL-12p40 transcriptional activity using *MAFB*-deficient 293T cell lines (Fig 1C and 1D). PU.1 synergistically activated the *Tnf* and *Il12b* promoters in MAFB knockout (KO) cells in the presence of NF-κB p65 and IRF-5 compared with wild type (WT) controls, indicating that the lack of MafB is responsible for the high sensitivity to these transcription factors (Fig 1C and 1D). Taken together, these data demonstrate that MafB is a negative regulator of the pro-inflammatory cytokines, TNF-α and IL-12p40.

### MafB interacts with both IRF-5 and PU.1

We subsequently performed a co-immunoprecipitation assay to elucidate the molecular mechanism of MafB-mediated inhibition (Fig 2A). An interaction of NF-κB p65 and IRF-5 has been previously reported [19], and it can be assumed that MafB interacts with the IRF family and E26 transformation-specific (Ets) family molecules, as demonstrated in several previous studies [13,14,16]. The results revealed that MafB interacts with both IRF-5, a member of the IRF family, and PU.1, a member of the Ets family, but not with NF-κB p65 (Fig 2A). A basic-leucine zipper (bZip) domain of MafB is known to act in a wide range of transcriptional regulation [14,20]. We generated a MafB bZip domain deletion (ΔbZip) mutant and analyzed the association and its activity in detail (Fig 2B–2D). The MafB ΔbZip mutant failed to bind to both IRF-5 and PU.1 (Fig 2B and 2C) and completely lost its inhibitory activity on transactivation of the TNF-α and IL-12p40 promoter (Fig 2D). These findings indicate that the bZip domain, possessed by MafB, is critical for MafB-mediated inhibition. Subsequently, we generated a domain deletion mutant of IRF-5 or PU.1 and performed a co-immunoprecipitation assay to identify the MafB bZip binding site (Fig 2E and 2F). Consistent with the results in Fig 2A, MafB interacted with the full-length IRF-5 and PU.1 but failed to interact with IRF-5 IRF association domain deletion (ΔIAD) mutant (Fig 2E) or PU.1 Ets domain deletion (ΔEts) mutant (Fig 2F), indicating that the IAD of IRF-5 and Ets domain of PU.1 are target sites for MafB bZip.

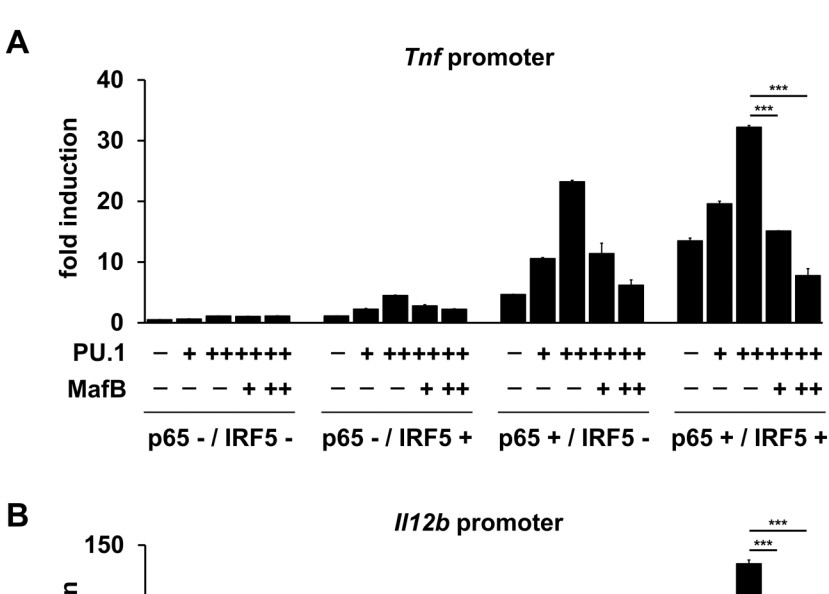

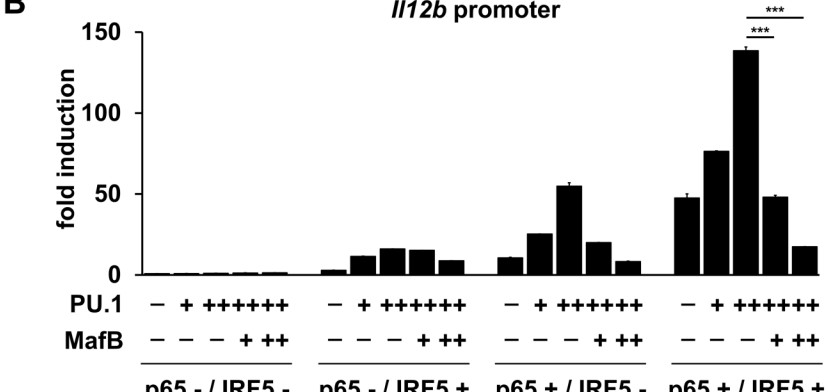

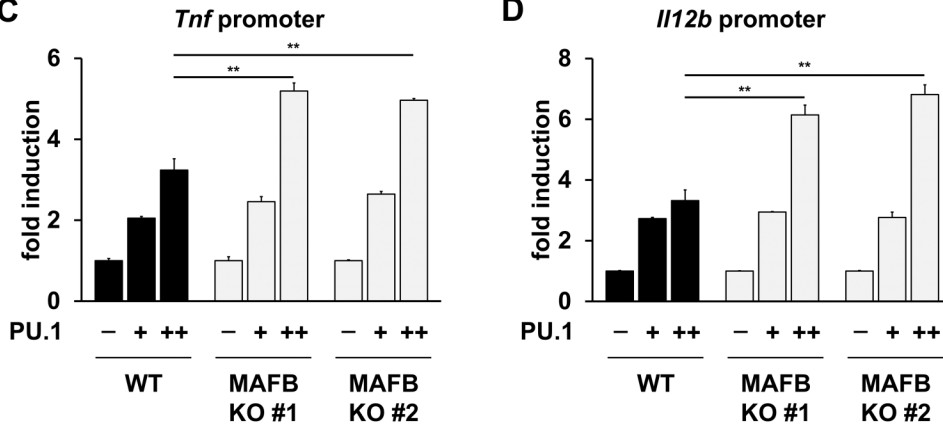

**Fig 1. Inhibition of TNF-α and IL-12p40 promoter activation by MafB. (A, B)** 293T cells were transfected with *Tnf* or *Il12b* promoter-driven luciferase reporter plasmid along with a combination of expression plasmids for p65 (-; 0 or +; 18 ng/well), IRF-5 (-; 0 or +; 91 ng/well), PU.1 (-; 0, +; 2.3, or ++; 9.1 ng/well), and/or MafB (-; 0, +; 5.7, or ++; 23 ng/well). After 24 h, cell lysates were measured using the luciferase assay kit. Luciferase activities are represented as a fold increase above background level of lysates prepared from mock-transfected cells. Data are representative of three independent experiments. ***$P < 0.001$; mean ± SD. **(C, D)** WT 293T cells or MAFB KO 293T cells were transfected with the *Tnf* or *Il12b* promoter-driven luciferase reporter plasmid along with a combination of expression plasmids for p65 (9 ng/well), IRF-5 (46 ng/well), and PU.1 (-; 0, +; 2.3, or ++; 9.1 ng/well). After 24 h, cell lysates were measured using the luciferase assay kit. Luciferase activities are represented as a fold increase above background level of lysates prepared from PU.1 negative cell. Data are representative of three independent experiments. **$P < 0.01$; mean ± SD.

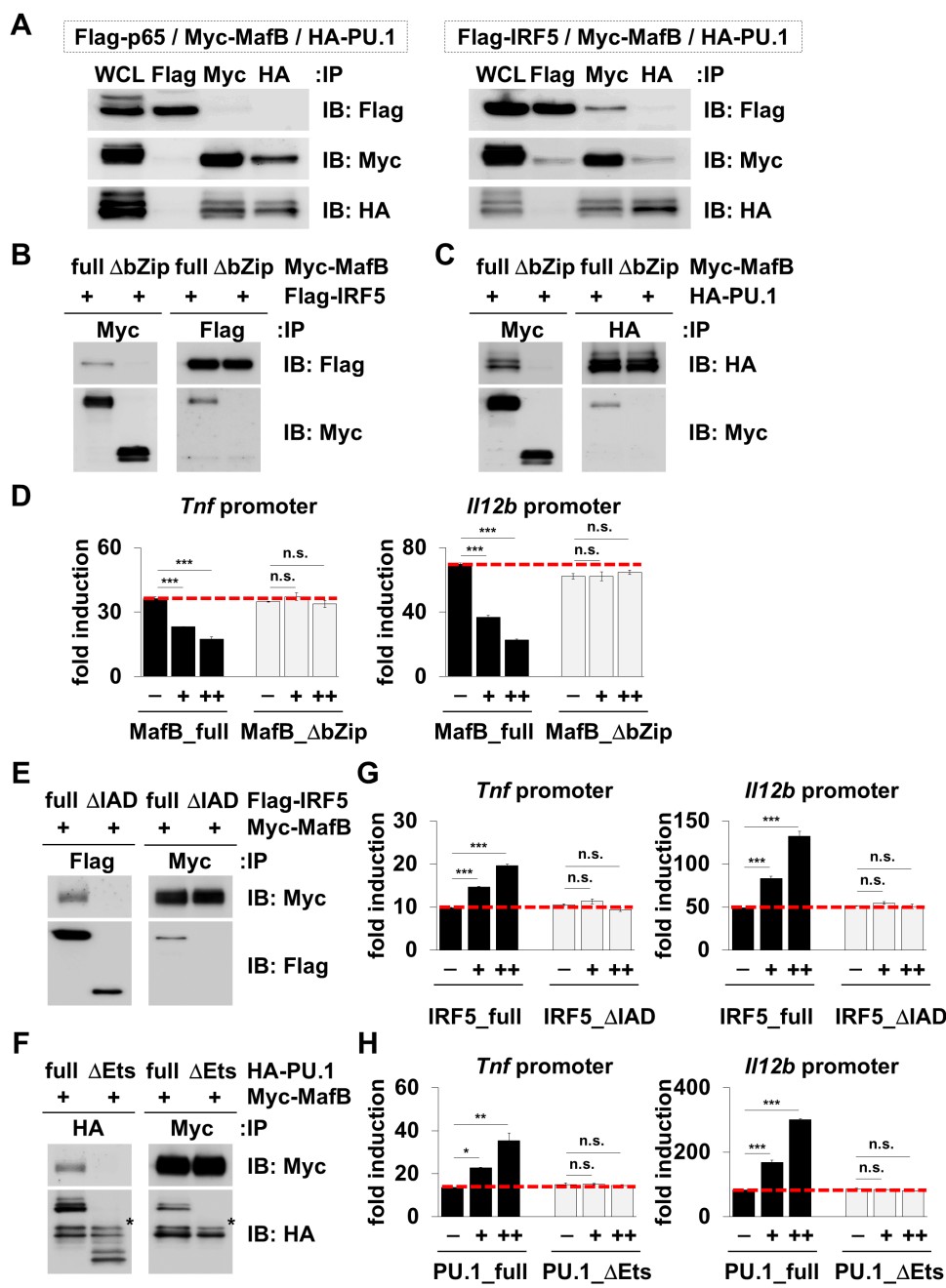

**Fig 2. Interaction of MafB with IRF-5 and PU.1. (A)** 293T cells were transiently transfected with the indicated expression plasmids using Lipofect-amine 2000. After 24 h, cell lysates were immunoprecipitated with the indicated antibodies, and tag-specific bands were subsequently detected using western blotting. Whole cell lysate (WCL) represents a positive control. Data are representative of three independent experiments. **(B, C)** 293T cells were transiently transfected with full-length Myc-tagged MafB (full), bZip domain deletion mutant (ΔbZip), and Flag-tagged IRF-5 (B) or HA-tagged PU.1 (C) expression plasmids using Lipofectamine 2000. After 24 h, cell lysates were immunoprecipitated with the indicated antibodies, and tag-specific bands were then detected using western blotting. Data are representative of three independent experiments. **(D)** 293T cells were transfected with *Tnf* or *Il12b* promoter-driven luciferase reporter plasmid along with a combination of expression plasmids for p65 (18 ng/well), IRF-5 (91 ng/well), PU.1 (9.1 ng/well), and/or MafB (-; 0, +; 2.9, or ++; 5.7 ng/well). After 24 h, cell lysates were measured using the luciferase assay kit. Luciferase activities are represented as a fold increase above background level of lysates prepared from mock-transfected cells. Data are representative of two independent experiments. ***$P<0.001$; n.s., not significant; mean ± SD. **(E, F)** 293T cells were transiently transfected with full-length Flag-tagged IRF-5 (full), IAD deletion mutant (ΔIAD), and Myc-tagged MafB (E) or full-length HA-tagged PU.1 (full), Ets domain deletion mutant (ΔEts), and Myc-tagged MafB (F) expression plasmids

using Lipofectamine 2000. After 24 h, cell lysates were immunoprecipitated with the indicated antibodies, and tag-specific bands were then detected using western blotting. Asterisks represent nonspecific bands. Data are representative of three independent experiments. **(G)** 293T cells were transfected with *Tnf* or *Il12b* promoter-driven luciferase reporter plasmid along with a combination of expression plasmids for p65 (18 ng/well) and/or IRF-5 (-; 0, +; 11, or ++; 45 ng/well). After 24 h, cell lysates were measured using the luciferase assay kit. Luciferase activities are represented as a fold increase above background level of lysates prepared from mock-transfected cells. Data are representative of two independent experiments. ***$P < 0.001$; n.s., not significant; mean ± SD. **(H)** 293T cells were transfected with *Tnf* or *Il12b* promoter-driven luciferase reporter plasmid along with a combination of expression plasmids for p65 (18 ng/well), IRF-5 (91 ng/well), and/or PU.1 (-; 0, +; 2.3, or ++; 9.1 ng/well). After 24 h, cell lysates were measured using the luciferase assay kit. Luciferase activities are represented as a fold increase above background level of lysates prepared from mock-transfected cells. Data are representative of two independent experiments. *$P < 0.05$; **$P < 0.01$; ***$P < 0.001$; n.s., not significant; mean ± SD.

Furthermore, none of the mutants exhibited TNF-α and IL-12p40 transcriptional activity (Fig 2G and 2H). This implies that the transcriptional activity of TNF-α and IL-12p40 requires the IAD and Ets domains of IRF5 and PU.1; MafB cancels this transcriptional activity via bZip by masking the IAD and Ets domains of the active site. We conclude that MafB does not have a suppressive function on its own but rather controls transcriptional activity by interfering with regions essential for transcriptional activity.

### MafB regulates TNF-α and IL-12p40 induction in macrophages

A previous study analyzing the expression pattern of MafB using various tissues, cell lines, and peritoneal macrophages reported that MafB is specifically expressed in macrophages [14]. We first assessed the localization of the related transcription factors containing MafB in macrophages to clarify the physiological role of MafB in the induction of TNF-α and IL-12p40 (Fig 3A). All transcription factors were detected within thioglycolate-elicited peritoneal macrophages, and both PU.1 and MafB were localized in the nucleus, whereas NF-κB p65, IRF-5, and part of MafB were localized in the cytosol (Fig 3A left). However, we found that a part of NF-κB p65 and IRF-5 resided in the nucleus and that MafB was in close proximity to these transcription factors under TLR stimulation (Fig 3A right), suggesting that all transcription factors share similar localization patterns in macrophages. Next, we investigated the expression of MafB in thioglycolate-elicited peritoneal macrophages stimulated with TLR ligands (Fig 3B and 3C). TLR2-induced activation led to an increase in the expression of several pro-inflammatory cytokines, including TNF-α and IL-12p40, as well as their related transcription factors. In contrast, MafB mRNA expression was sequentially reduced after priming with TLR2 ligands (Fig 3B). Furthermore, immunoblot analysis using TLR2-stimulated macrophages revealed that the MafB protein was degraded in a time-dependent manner and that treatment with proteasome inhibitors such as MG-132 and bortezomib canceled the degradation of MafB during the early phase of immune responses (Fig 3C). These results suggest that lower levels of MafB expression are required for the induction of TNF-α and IL-12p40 in macrophages. Subsequently, we also analyzed TNF-α and IL-12p40 production by thioglycolate-elicited peritoneal macrophages treated with all-trans retinoic acid (RA), known to elevate the expression of MafB in macrophages [21] to test whether MafB mRNA and protein expression levels control the induction of TNF-α and IL-12p40 in macrophages. MafB mRNA expression increased in thioglycolate-elicited peritoneal macrophages in a dose-dependent manner (Fig 3D). Additionally, RA treatment led to an increase in the MafB protein abundance compared with untreated cells (Fig 3E). Under similar conditions, we found that high MafB expression suppressed TNF-α and IL-12p40 production (Fig 3F). Furthermore, it was evident that *Tnf* and *Il12b* fully correlated with *Mafb* (Fig 3G), supporting our view of the molecular mechanism as shown in Fig 2. In conclusion, these findings point to MafB as a regulator of TNF-α and IL-12p40 induction in macrophages.

### *M. tuberculosis* controls cytokine production via MafB

Meta-analyses of GWAS have reported increasing expression of MAFB in patients with active TB [6,7], and a recent study suggested an association between MAFB and *M. tuberculosis* infection in human monocyte cell lines [22]. However, the details of the role of transcription factor MafB in *M. tuberculosis* infection remain unknown. We first analyzed

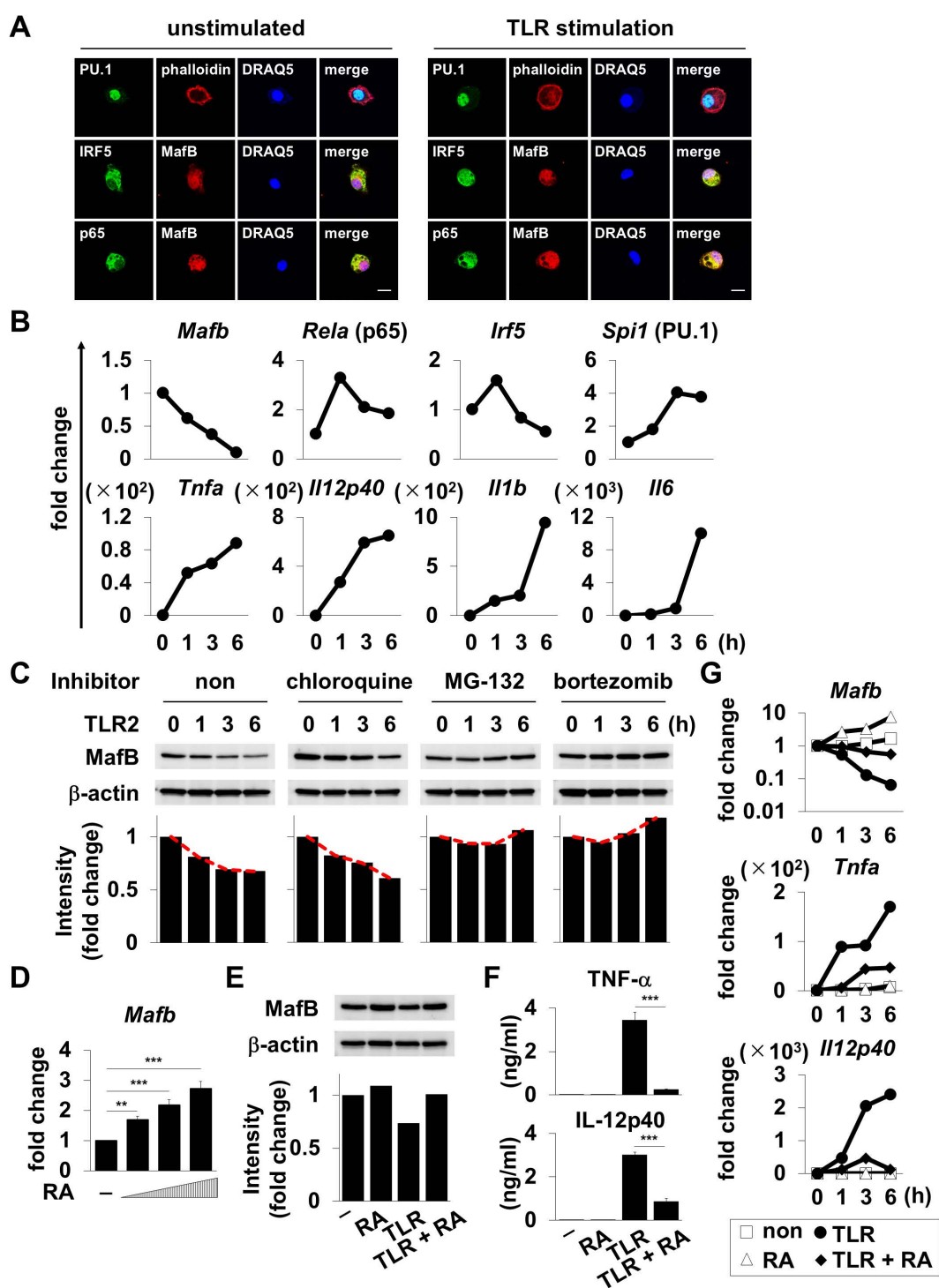

**Fig 3. Regulation of TNF-α and IL-12p40 by MafB. (A)** Thioglycolate-elicited peritoneal macrophages were stimulated with or without 100 ng/mL Pam3CSK4. After 3 h, cells were stained with antibodies to PU.1, phalloidin, and nucleus (upper panels); IRF-5, MafB, and nucleus (middle panels); or p65, MafB, and nucleus (lower panels) and visualized using fluorescence microscopy. Scale bar represents 10 μm. **(B)** Thioglycolate-elicited peritoneal macrophages were stimulated with 100 ng/mL Pam3CSK4. Cells were harvested at the indicated time points and analyzed using quantitative RT–PCR. Relative gene expression is represented as a fold increase above background level of samples prepared from unstimulated cells. Data are representative of three independent experiments. **(C)** Thioglycolate-elicited peritoneal macrophages were treated with 5 μg/mL cycloheximide to inhibit protein

synthesis, and cells were then stimulated with 100 ng/mL Pam3CSK4 after culture in the presence or absence of 100 μM chloroquine, 100 nM MG-132, or 100 nM bortezomib. Cells were harvested at the indicated time points, and MafB-specific bands were detected using western blotting. Anti-β-actin bands represent a positive control for the cell lysate. The intensity of MafB-specific bands is quantified using the ImageQuant TL analysis software. Data are representative of two independent experiments. **(D)** Thioglycolate-elicited peritoneal macrophages were treated with 0, 1, 5, or 10 μM RA. After 6 h, cells were harvested and analyzed using quantitative RT–PCR. *Mafb* expression was represented as a fold increase above background level of samples prepared from untreated cells. Data are representative of two independent experiments. **P < 0.01; ***P < 0.001; mean ± SD. **(E)** Thioglycolate-elicited peritoneal macrophages were stimulated with or without 100 ng/mL Pam3CSK4 after culture in the presence or absence of 30 μM RA. Cells were harvested after 6 h, and MafB-specific bands were detected using western blotting. Anti-β-actin bands represent a positive control for the cell lysate. The intensity of MafB-specific bands is quantified using the ImageQuant TL analysis software. Data are representative of three independent experiments. **(F)** Thioglycolate-elicited peritoneal macrophages were stimulated with or without 100 ng/mL Pam3CSK4 after priming with or without 30 μM RA. After 24 h, cytokine production was measured using ELISA. Data are representative of three independent experiments. ***P < 0.001; mean ± SD. **(G)** Thioglycolate-elicited peritoneal macrophages were stimulated with or without 100 ng/mL Pam3CSK4 after priming with or without 30 μM RA. Cells were harvested at the indicated time points and analyzed using quantitative RT–PCR. Relative gene expression is represented as a fold increase above background level of samples prepared from unstimulated cells. Data are representative of three independent experiments.

MafB expression and cytokine production in macrophages after priming with TLR2 ligands known as cell wall components found in mycobacteria including *M. tuberculosis*. Stimulation with TLR2 ligands decreased MafB mRNA expression in a dose-dependent manner, along with the expression level of MafB protein in thioglycolate-elicited peritoneal macrophages (Fig 4A and 4B). Consequently, TNF-α and IL-12p40 production was elevated (Fig 4C). In contrast, a bacterial load-dependent increase was observed in MafB expression along with poor cytokine production (Fig 4D–4F). These findings suggest that MafB regulates cytokine production even during *M. tuberculosis* infection and raise questions as to whether *M. tuberculosis* may prevent the inherent ability of macrophages to control pathogens via MafB. We carried out subsequent experiments using macrophage-specific *Mafb*-deficient (*Mafb*f/f::LysM-Cre) mice to address this question (Figs 4G–4K, S1, and S2 Figs). *M. tuberculosis* infection led to an increase in the mRNA expression of all pro-inflammatory cytokine genes in thioglycolate-elicited peritoneal macrophages (Fig 4G). Particularly, mRNA levels of *Tnf* and *Il12b* were higher in the MafB-conditional knockout (cKO) macrophages than in control (*Mafb*f/f) macrophages. Furthermore, *Mafb*-deficient macrophages secreted a large amount of these cytokines upon infection with *M. tuberculosis* (Fig 4H), indicating that MafB limits the secretion of TNF-α and IL-12p40 in *M. tuberculosis*-infected macrophages. Next, we performed a colony-forming unit (CFU) assay to assess the impact of MafB-mediated cytokine restriction on *M. tuberculosis* survival. The CFU titers in both groups were comparable in the early phase of infection; however, a profound difference in the numbers of *M. tuberculosis* were observed 24 h after infection, with a gradual decline in the CFU titer in *Mafb*-deficient macrophages (Fig 4I). TNF-α and IL-12p40 are well-known to activate macrophages, leading to the production of nitric oxide intermediates (NOI), the master mediators of host defense against intracellular pathogens including *M. tuberculosis* [23–26]. We therefore determined NO production in thioglycolate-elicited peritoneal macrophages (Fig 4J and 4K). The transcriptional levels of Nos2 and other bactericidal inflammatory effectors were markedly increased in *M. tuberculosis*-infected *Mafb*-deficient macrophages (Figs 4J and S3), with a corresponding elevation in NO levels (Fig 4K). Finally, we attempted to identify the causative factors responsible for the elevated MafB expression in *M. tuberculosis* infection. Previous studies have reported that anti-inflammatory cytokines, such as IL-10, contribute to elevated MafB expression [27,28]. Consistently, exogenous recombinant IL-10 upregulated MafB expression (Fig 4L). Furthermore, *Il10* expression was markedly elevated in thioglycolate-elicited peritoneal macrophages infected with a high-dose of *M. tuberculosis* (Fig 4M). Meanwhile, inhibition of the IL-10 receptor abrogated the *M. tuberculosis*-induced increase in MafB expression (Figs 4N and S4), suggesting that *M. tuberculosis*-inducible IL-10 is responsible for upregulating MafB. Taken together, these findings indicate that *M. tuberculosis* takes advantage of MafB to facilitate its survival in macrophages.

## MafB contributes to exacerbated *M. tuberculosis* infection

We performed *in vivo* experiments using macrophage-specific *Mafb*-deficient mice to determine the biological role of the transcription factor MafB in *M. tuberculosis* infection. MafB mRNA expression was enhanced in the lungs of control mice

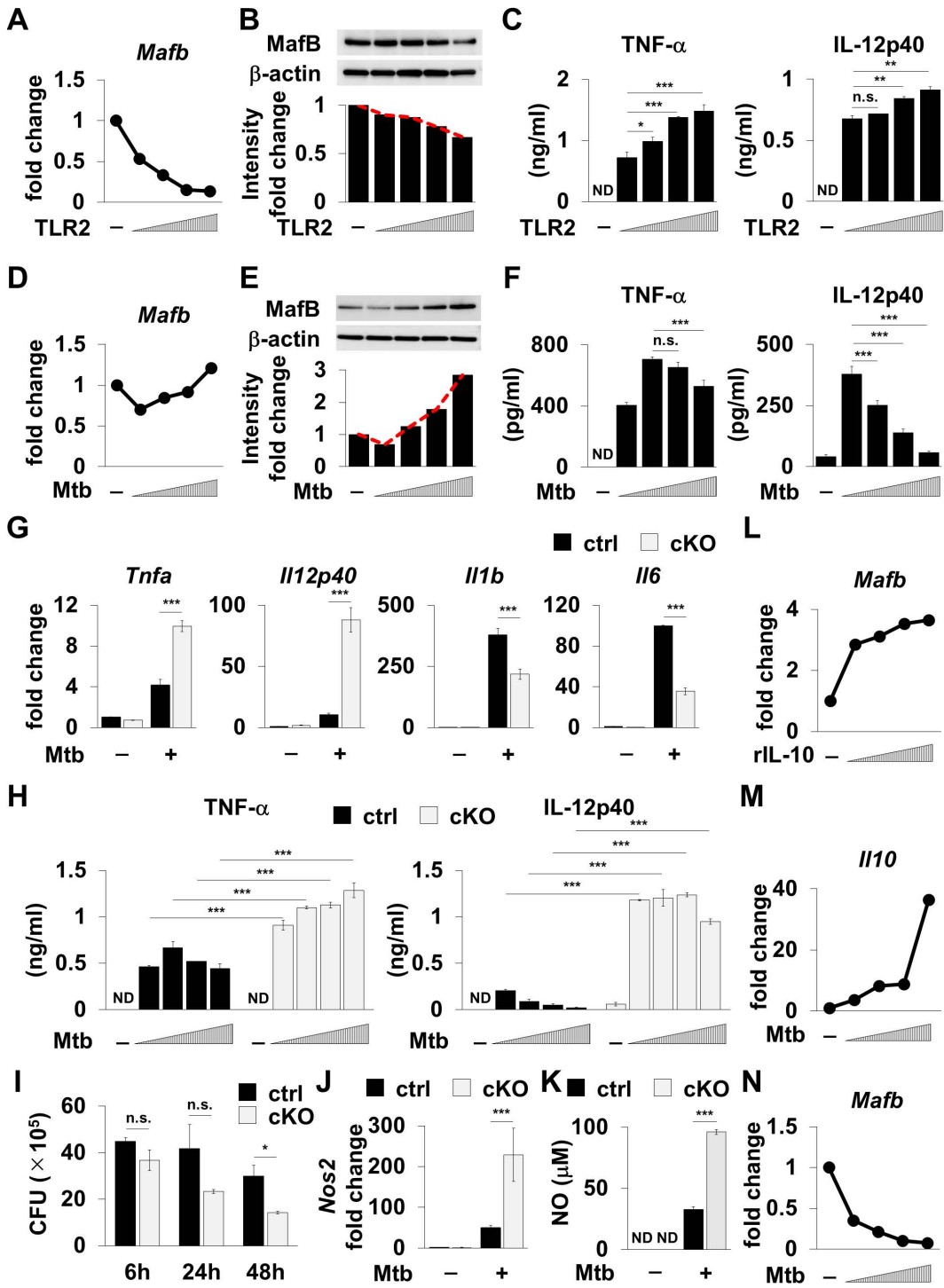

**Fig 4. MafB-dependent cytokine restriction in *M. tuberculosis* infection. (A)** Thioglycolate-elicited peritoneal macrophages were stimulated with 0, 100, 200, 400, or 500 ng/mL Pam3CSK4. After 6 h, cells were harvested and analyzed using quantitative RT–PCR. *Mafb* expression is represented as a fold increase above background level of samples prepared from untreated cells. Data are representative of two independent experiments. **(B)** Thioglycolate-elicited peritoneal macrophages were stimulated with 0, 100, 200, 400, or 500 ng/mL Pam3CSK4. Cells were harvested after 6 h, and MafB-specific bands were detected using western blotting. Anti-β-actin bands represent a positive control for the cell lysate. The intensity of MafB-specific bands is quantified using the ImageQuant TL analysis software. Data are representative of three independent experiments. **(C)**

Thioglycolate-elicited peritoneal macrophages were stimulated with 0, 100, 200, 400, or 500 ng/mL Pam3CSK4. After 6 h, cytokine production was measured using ELISA. Data are representative of three independent experiments. *$P<0.05$; **$P<0.01$; ***$P<0.001$; ND, not detected; n.s., not significant; mean ± SD. **(D)** Thioglycolate-elicited peritoneal macrophages were infected with *M. tuberculosis* H37Rv (MOI of 0, 0.5, 1, 2, or 5). After 6 h, cells were harvested and analyzed using quantitative RT–PCR. *Mafb* expression is represented as a fold increase above background level of samples prepared from uninfected cells. Data are representative of three independent experiments. **(E)** Thioglycolate-elicited peritoneal macrophages were infected with *M. tuberculosis* H37Rv (MOI of 0, 1, 2, 3, or 4). Cells were harvested after 6 h, and MafB-specific bands were detected using western blotting. Anti-β-actin bands represent a positive control for the cell lysate. The intensity of MafB-specific bands is quantified using the ImageQuant TL analysis software. Data are representative of two independent experiments. **(F)** Thioglycolate-elicited peritoneal macrophages were infected with *M. tuberculosis* H37Rv (MOI of 0, 1, 2, 3, or 4). After 24 h, cytokine production was measured using ELISA. Data are representative of three independent experiments. ***$P<0.001$; ND, not detected; n.s., not significant; mean ± SD. **(G)** Thioglycolate-elicited peritoneal macrophages were infected with or without *M. tuberculosis* H37Rv (MOI of 5). After 6 h, cells were harvested and analyzed using quantitative RT–PCR. Relative gene expression is represented as a fold increase above background level of samples prepared from uninfected control cells. Data are representative of three independent experiments. ***$P<0.001$; mean ± SD. **(H)** Thioglycolate-elicited peritoneal macrophages were infected with *M. tuberculosis* H37Rv (MOI of 0, 1, 2, 3, or 4). After 24 h, cytokine production was measured using ELISA. Data are representative of three independent experiments. ***$P<0.001$; ND, not detected; mean ± SD. **(I)** Thioglycolate-elicited peritoneal macrophages were infected with *M. tuberculosis* H37Rv (MOI of 5). At the indicated time points after infection, cells were harvested and plated onto 7H10-OADC agar. Data are representative of three independent experiments. *$P<0.05$; n.s., not significant; mean ± SD. **(J)** Thioglycolate-elicited peritoneal macrophages were infected with or without *M. tuberculosis* H37Rv (MOI of 5). After 6 h, cells were harvested and analyzed using quantitative RT–PCR. *Nos2* expression is represented as a fold increase above background level of samples prepared from uninfected cells. Data are representative of three independent experiments. ***$P<0.001$; mean ± SD. **(K)** Thioglycolate-elicited peritoneal macrophages were infected with or without *M. tuberculosis* H37Rv (MOI of 5). After 24 h, NO levels in the culture supernatants were measured. Data are representative of three independent experiments. ***$P<0.001$; ND, not detected; mean ± SD. **(L)** Thioglycolate-elicited peritoneal macrophages were stimulated with 0, 10, 20, 50, or 100 ng/mL recombinant IL-10. After 6 h, cells were harvested and analyzed using quantitative RT–PCR. *Mafb* expression is represented as a fold increase above background level of samples prepared from untreated cells. Data are representative of two independent experiments. **(M)** Thioglycolate-elicited peritoneal macrophages were infected with *M. tuberculosis* H37Rv (MOI of 0, 0.5, 1, 2, or 5). After 6 h, cells were harvested and analyzed using quantitative RT–PCR. *Il10* expression is represented as a fold increase above background level of samples prepared from uninfected cells. Data are representative of two independent experiments. **(N)** Thioglycolate-elicited peritoneal macrophages were infected with *M. tuberculosis* H37Rv (MOI of 0, 0.5, 1, 2, or 5) after priming with 100 µg/mL anti-mouse IL-10R. After 6 h, cells were harvested and analyzed using quantitative RT–PCR. *Mafb* expression is represented as a fold increase above background level of samples prepared from untreated cells. Data are representative of two independent experiments.

in the active phase of TB compared with that in uninfected mice, whereas MafB expression was considerably lower in MafB cKO lungs (Fig 5A), indicating that MafB expression in lungs is dependent on tissue macrophages. Having established that macrophages are the main source of MafB during *M. tuberculosis* infection, we next ascertained whether MafB-dependent cytokine restriction also occurs in experimental TB in mice (Fig 5B and 5C). In lungs infected with *M. tuberculosis*, TNF-α and IL-12p40 mRNA expression was higher in *Mafb*-deficient mice (Fig 5B). At the protein level, TNF-α concentration in the serum was low and not significantly different during the active phases of TB (Fig 5C left), while secretion levels of IL-12p40 were upregulated owing to the lack of MafB in *Mafb*-deficient mice (Fig 5C right). These findings demonstrate that *M. tuberculosis* limits cytokine production by enhancing MafB expression *in vivo*, although it remains to be clarified how the elevated MafB expression affects the survival of mice infected with *M. tuberculosis* and bacterial numbers within their organs. Therefore, we further evaluated the survival rate and CFU numbers of mice infected with *M. tuberculosis* (Fig 5D–5G). Control mice and macrophage-specific *Mafb*-deficient mice were intratracheally infected with the virulent *M. tuberculosis* H37Rv strain and monitored for their survival (Fig 5D). Control mice harboring MafB began to die 4 weeks after infection, and several *M. tuberculosis*-infected mice died by 10 weeks. In contrast, almost all *Mafb*-deficient mice survived at least 10 weeks post-infection. Computed tomography (CT) scans to define disease severity corroborated the milder spread of TB foci and resistance to *M. tuberculosis* infection in *Mafb*-deficient mice (Fig 5E). Furthermore, we identified characteristic histological features of TB, including several changes in granulomas harboring acid-fast bacilli and cluster assembly of macrophages by histopathological analysis in control mice. Additionally, increased disease severity in the lungs of control mice was evident at autopsy with signs of pulmonary lesions (Figs 5F and S5). In contrast, CFU titers of *M. tuberculosis* were lower in all organs in *Mafb*-deficient mice (Fig 5G). These results demonstrate that *Mafb*-deficient mice are highly resistant to *M. tuberculosis* infection. Finally, we assessed the survival of mice with *M. tuberculosis* infection after inoculation of neutralizing monoclonal antibodies to determine whether enhanced cytokine production in *Mafb*-deficient mice is responsible for high resistance to *M. tuberculosis* infection (Fig 5H). Antibody

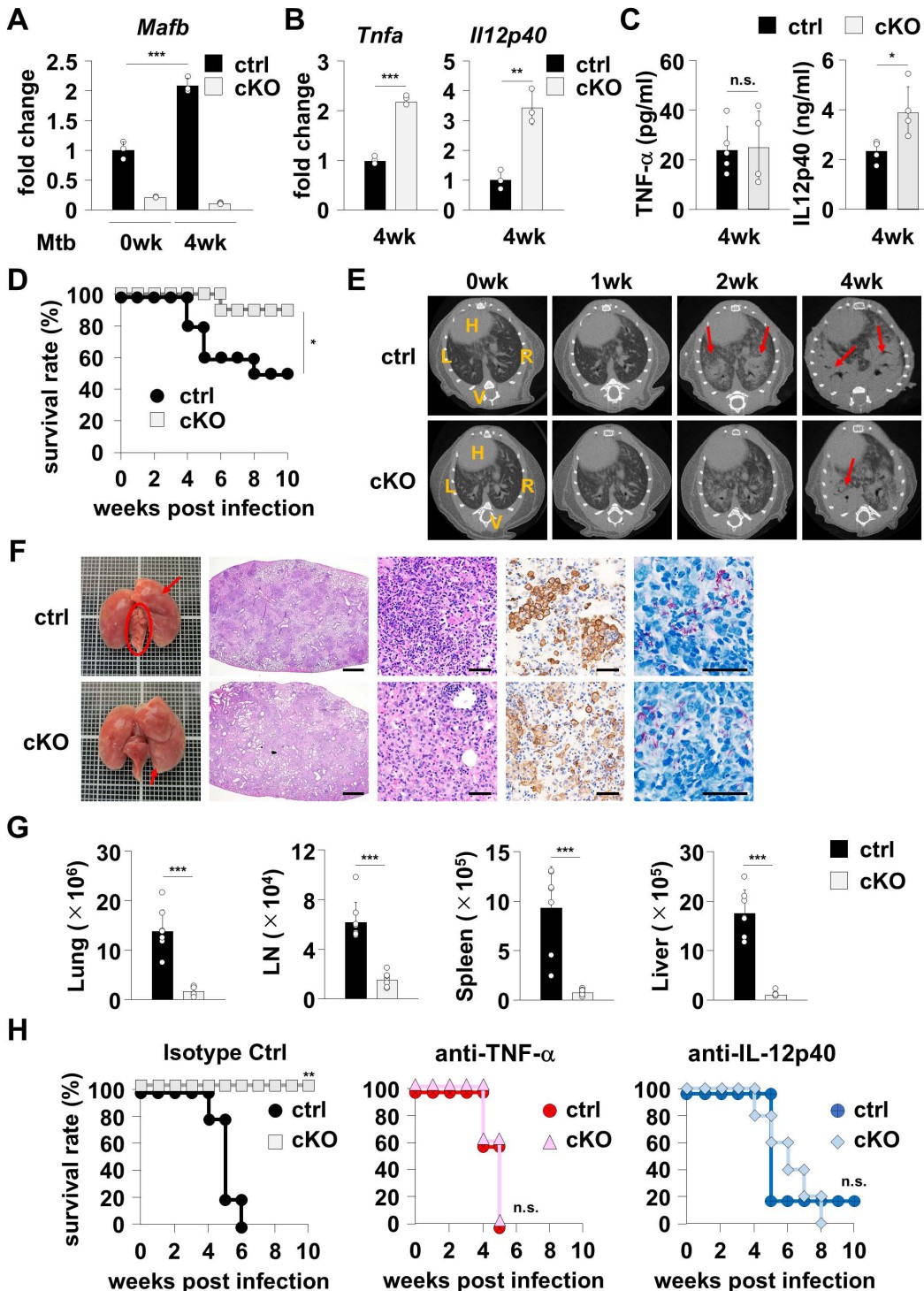

**Fig 5. High resistance to *M. tuberculosis* infection in *Mafb*-deficient mice. (A)** *Mafb*f/f (n = 3 mice per group) and *Mafb*f/f::LysM-Cre (n = 3 mice per group) mice were intratracheally infected with *M. tuberculosis* H37Rv (5 × 10³ CFU). Lungs were homogenized at the indicated time points and analyzed using quantitative RT–PCR. *Mafb* expression is represented as a fold increase above background level of samples prepared from uninfected control mice. Data are representative of two independent experiments. ***P < 0.001; mean ± SD. **(B)** *Mafb*f/f (n = 3 mice per group) and *Mafb*f/f::LysM-Cre (n = 3 mice per group) mice were intratracheally infected with *M. tuberculosis* H37Rv (5 × 10³ CFU). At 4 weeks after infection, lungs were homogenized and analyzed using quantitative RT–PCR. mRNA expression is represented as a fold increase above background level of samples prepared from control

mice. Data are representative of two independent experiments. **$P < 0.01$; ***$P < 0.001$; mean ± SD. **(C)** *Mafb*^f/f (n = 5 mice per group) and *Mafb*^f/f::LysM-Cre (n = 5 mice per group) mice were intratracheally infected with *M. tuberculosis* H37Rv ($5 \times 10^3$ CFU). Blood samples were collected at 4 weeks post infection, and cytokine production was measured using ELISA. Data are representative of two independent experiments. *$P < 0.05$; n.s., not significant; mean ± SD. **(D)** *Mafb*^f/f (n = 10) and *Mafb*^f/f::LysM-Cre (n = 10) mice were intratracheally infected with *M. tuberculosis* H37Rv ($5 \times 10^3$ CFU), and their survival was monitored for 10 weeks. Data are representative of three independent experiments. *$P < 0.05$. **(E)** *Mafb*^f/f (n = 10) and *Mafb*^f/f::LysM-Cre (n = 10) mice were intratracheally infected with *M. tuberculosis* H37Rv ($5 \times 10^3$ CFU). At 0, 1, 2, or 4 weeks after infection, lungs were digitized by micro-CT scan. Red arrows represent several pulmonary lesions. H, heart; V, vertebral column; L, left lung; R, right lung. **(F)** *Mafb*^f/f (n = 3) and *Mafb*^f/f::LysM-Cre (n = 3) mice were intratracheally infected with *M. tuberculosis* H37Rv ($5 \times 10^3$ CFU). Three weeks after infection, lungs were embedded in paraffin, and lung tissue sections were used for hematoxylin-eosin staining (Scale bar represents 1 mm or 50 μm), Iba1 immunohistochemical staining (Scale bar represents 50 μm), and Ziehl-Neelsen staining (Scale bar represents 50 μm). Red arrows and frames represent several pulmonary granulomatous lesions. **(G)** *Mafb*^f/f (n = 7) and *Mafb*^f/f::LysM-Cre (n = 7) mice were intratracheally infected with *M. tuberculosis* H37Rv ($5 \times 10^3$ CFU). Three weeks after infection, homogenates of lungs, mediastinal lymph nodes, spleens, and livers were plated on 7H10-OADC agar, and the CFU titers were counted. Data are representative of three independent experiments. ***$P < 0.001$; mean ± SD. **(H)** *Mafb*^f/f (n = 5 mice per group) and *Mafb*^f/f::LysM-Cre (n = 5 mice per group) mice were intratracheally infected with *M. tuberculosis* H37Rv ($5 \times 10^3$ CFU) and their survival was monitored for 10 weeks. Neutralizing monoclonal antibodies (200 μg/ 250 μL/ mouse) were inoculated into the abdominal cavity of mice every other day until 4 weeks of infection. Data are representative of two independent experiments, and significant differences are compared between control and cKO mice. **$P < 0.01$; n.s., not significant.

neutralization of the bioactivity of natural TNF-α or IL-12p40 resulted in the loss of protection against *M. tuberculosis* in *Mafb*-deficient mice (Fig 5H middle and right), demonstrating that the high resistance of *Mafb*-deficient mice to *M. tuberculosis* infection is attributable to elevated production of TNF-α and IL-12p40. Overall, MafB-mediated cytokine restrictions by *M. tuberculosis* could serve as a valuable target for future host-directed therapies.

## Discussion

MafB is highly expressed in macrophages; however, its corresponding role remains poorly understood. The present study demonstrates that the transcription factor MafB is a regulator for the induction of pro-inflammatory cytokines TNF-α and IL-12p40. Specifically, our data demonstrate that reduced MafB expression is required for cytokine production in response to TLR stimulation, whereas enhanced MafB expression leads to poor cytokine production during *M. tuberculosis* infection. Thus, the transcription factor MafB appears to be critically involved in pro-inflammatory cytokine induction in macrophages.

Detailed molecular mechanistic analysis revealed that the interference of MafB with IRF-5 and PU.1, major components involved in cytokine induction, is key to MafB-mediated cytokine regulation, although direct effects of MafB on NF-κB p65 need to be analyzed in the future (Fig 6). Because these components control the transcriptional activity of several inflammatory effectors, including TNF-α and IL-12p40 [19], MafB may also affect the expression of other inflammatory genes. In this context, we found that not only *Tnf* and *Il12b* but also *Lcn2* and *Slpi* expressions were elevated in cKO macrophages. The effect of MafB on the expression of inflammatory genes in macrophages and in animal models has been described previously [29–31]. This implies that MafB plays a pivotal role in broadly regulating induction of inflammatory effectors in macrophages. Furthermore, our immunoblot analyses demonstrate that MafB directly binds to IRF-5 and PU.1, members of the IRF family and Ets family, respectively. Additionally, our previous experiments show that MafB binds to other family members involved in type I IFN production in pDC, IRF-7, and Spi-B [13]. This is consistent with several studies that report that MafB associates with other IRF and Ets families [14,16,18]. Thus, MafB potentially contributes to diverse intracellular activities involving IRF and Ets family cognates involved in various autoimmune and infectious diseases. Nevertheless, the HEK293T cells-based system findings lack physiological relevance, including cell types and cell activity states, and must be interpreted with caution. Although analyzing intracellular interactions in primary macrophages and pDCs is the most reliable method to overcome these issues, the use of primary cells presents its own challenges, such as 1) condition of target antibodies, 2) difficulty in cell harvest, and 3) low transfection efficiency. Indeed, replicating the results obtained using 293T cells has proven difficult. Advances in technology that enhance transfection efficiency in primary cells and improve antibody quality are therefore highly anticipated.

**steady state**

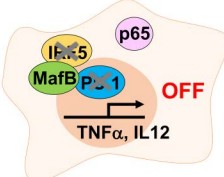

**TLR stimulation**

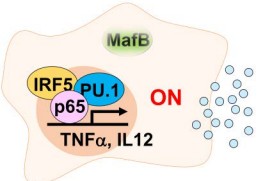

*M. tuberculosis* infection

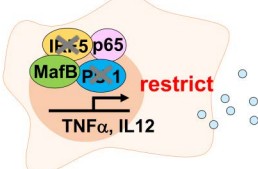

**Fig 6. MafB-dependent cytokine regulation in macrophages.** Schematic model of pro-inflammatory cytokine regulation by the transcription factor MafB. Under steady-state conditions or *M. tuberculosis* infection, MafB abrogates the induction of pro-inflammatory cytokines by interfering with IRF-5 and PU.1, which are key components essential for transcriptional activity. Upon stimulation by TLR ligands, the transcription and protein expression levels of MafB are sequentially reduced in macrophages, thereby leading to an abundance of cytokine production.

The cause of increased MafB expression in TB remains unclear. Our previous study on pDCs showed that the TLR7 or TLR9 ligands decreased MafB mRNA and protein expression levels [13]. Here, we describe similar results for macrophages in response to TLR2 stimulation with bacterial lipopeptides of mycobacteria. A reduction in MafB expression in bone marrow-derived macrophages primed by the TLR4 ligand, LPS, has also been reported [27]. Thus, signaling via TLRs negatively regulates MafB. As demonstrated prior, a reduction in MafB expression under low *M. tuberculosis* burden may be due to TLR signaling. In contrast, we believe that *M. tuberculosis*-dependent IL-10 is a strong candidate for direct triggering of MafB expression. *M. tuberculosis* infection in thioglycolate-elicited peritoneal macrophages resulted in increased *Il10* gene expression depending on the bacterial load; high MafB expression was dependent on the dose of exogenous IL-10. These results are consistent with previous studies, which revealed that the anti-inflammatory cytokines IL-10, IL-4, and IL-13 contribute to elevated MafB expression [27,28] and MafB is a target gene of the IL-10/STAT3 signaling pathway [32]. In addition, anti-IL-10R treatment abrogated the elevated MafB expression. Thus, the impact of *M. tuberculosis*-inducible IL-10 on MafB could be predominant over that of TLR signaling under high-dose *M. tuberculosis* infection. Interestingly, a recent study demonstrated that inhibition of triggering receptor expressed on myeloid cells 2 (Trem2), which mediates anti-inflammatory immune signaling, reduces microglial populations with high MafB expression [33]. Other studies have reported that Trem2 contributes to *M. tuberculosis* survival by promoting IL-10 production, suppressing pro-inflammatory cytokine secretion, and limiting macrophage activation against mycobacteria in a Trem2-dependent manner [34,35]. Consistent with these findings and our current data, we propose that Trem2-mediated signaling, including IL-10 induction, may be responsible for upregulating MafB expression during *M. tuberculosis* infection. Alternatively, nuclear receptors could promote MafB expression. RA induces MafB expression via the retinoic acid receptor (RAR), a member of the nuclear receptor family [21]. Consistently, a recent study using a pharmacological inhibitor found in mice demonstrated that *M. tuberculosis* activates the endogenous RA/RAR pathway in murine monocytes and macrophages to undermine host protection against *M. tuberculosis* [36]. Moreover, an increased abundance of peroxisome proliferator-activated receptor (PPAR)-γ, another nuclear receptor known as MafB inducing factor [37], in *M. tuberculosis*-infected macrophages has been associated with the accumulation of lipid droplets and formation of foamy macrophages, favoring growth and persistence

of *M. tuberculosis* [38,39]. Indeed, inhibition of these receptors further abrogated the induction of MafB expression by *M. tuberculosis*. In combination, these factors could bring about high MafB expression during *M. tuberculosis* infection.

Because *Mafb*-deficient mice die within 24 h after birth due to cyanosis and fatal central apnea [10,40], it is difficult to analyze MafB function in adult mice; therefore, we performed *in vivo* experiments using macrophage-specific *Mafb*-deficient mice to overcome these limitations in analyzing the role of MafB in TB. Consistent with reports on patients with TB, we detected high levels of MafB in the lungs of control mice during the active phase of infection, along with poorer IL-12p40 production as compared to cKO mice during early phase of infection, i.e., cytokine restrictions by mycobacteria via MafB. This underlines the validity of our *in vitro* experiments. Furthermore, MafB cKO mice showed higher resistance to *M. tuberculosis* infection than control mice at higher doses of infection. The critical role of TNF-α and IL-12p40 in host defense against *M. tuberculosis* infection has been demonstrated by previous studies using gene-deficient and transgenic mice and in a recent article on patients with TNF deficiency [23,41–43]. Consistent with these studies, our data revealed ameliorated TB lesions and lower *M. tuberculosis* load in the lungs of cKO mice. Additionally, our experiments using neutralizing antibodies support the role of elevated TNF-α and IL-12p40 production in TB resistance in cKO mice. Various rare human diseases caused by MafB variants have been successfully modeled in mice carrying analogous *Mafb* mutations, which recapitulate similar symptoms and key disease features, providing a strong suggestion of a correlation between mice and humans [44,45]. Based on our current mouse studies, we conclude that elevated MAFB expression in patients with active TB likely promotes the persistence of *M. tuberculosis,* as observed in mice. Histopathological analysis of murine lungs after *M. tuberculosis* infection revealed notable differences between control and *Mafb*-deficient macrophages. As demonstrated earlier, no significant difference was observed in the number of Iba1-positive cells, known as the macrophage-specific protein, between the two groups; however, control mice presented several clusters of large cells with a darkly stained cell surface. This is similar to the characteristics of foamy macrophages [46,47], which serve as nutrient-rich reservoirs for *M. tuberculosis* persistence. In addition, we found considerably more Arginase-1-positive cells in the same sections of control mice and fewer iNOS-positive cells than those in cKO mice. These are characteristic markers of M2 (anti-inflammatory state) and M1 (pro-inflammatory state) macrophages, respectively. Additionally, foamy macrophages are known to show the M2-like phenotype [47], indicating that M2 macrophages are prominent in MafB-possessing mice during *M. tuberculosis* infection. This is consistent with a previous study that revealed that MafB promotes macrophage phenotype into the M2 state by directly enhancing the expression of M2 polarization marker genes [15]. Thus, *M. tuberculosis* promotes a shift to M2 macrophages over M1 macrophages by upregulating MafB expression. This shift in turn creates a hospitable environment for *M. tuberculosis*. Further studies will help elucidate the role of MafB in infectious diseases and may lay the groundwork for novel host-directed therapies against TB and other infections [48].

## Conclusion

Our findings provide novel mechanistic insights into the survival stratagem of *M. tuberculosis* harnessing MafB by demonstrating that MafB regulates the induction of pro-inflammatory cytokines TNF-α and IL-12p40 in macrophages. Our study highlights the pivotal role of MafB as a biomarker for TB susceptibility.

## Materials and methods

### Ethics statement

All animal experiments were approved by and conducted in accordance with the guidelines of the Animal Use and Care Committee of the National Institute of Infectious Diseases.

### Animals

C57BL/6 mice (8–10 weeks old) were purchased from CLEA Japan. *Mafb*^f/f and *Mafb*^f/f::LysM-Cre mice were provided [18]. *Mafb*^f/f and *Mafb*^f/f::LysM-Cre littermates from intercrosses between male *Mafb*^f/f::LysM-Cre mice and female *Mafb*^f/f mice aged 8–10 weeks were used for experiments. The mice were bred and maintained under specific pathogen-free conditions.

## Cells

Human embryonic kidney cells with a gene encoding the SV40 large T antigen (HEK293T cells) were cultured in DMEM with high glucose (Nacalai Tesque, Kyoto, Japan) containing 10% (vol/vol) heat-inactivated fetal bovine serum (FBS). *MAFB*-deficient 293T cell lines were generated as described previously [13]. Briefly, the designed insert fragments were subcloned into the CRISPR/Cas9 expression vector pSpCas9(BB)-2A-Puro (PX459) V2.0. Subsequently, 293T cells were transfected with these plasmids using ScreenFect A (Fujifilm Wako Pure Chemical, Osaka, Japan) and cultured in DMEM containing 1 µg/mL puromycin (InvivoGen, San Diego, CA, USA) for two days. Then, the puromycin-resistant cells were isolated. The *MAFB*-specific mutations were confirmed using sequence analysis, and the two cell lines, 1–100-B9 and 3–100-A11, were used for luciferase assay. Mice were intraperitoneally injected with 4% thioglycolate. Macrophages were isolated from the peritoneal cavity three days later and cultured in RPMI 1640 medium (Nacalai Tesque) containing 10% (vol/vol) heat-inactivated FBS to obtain peritoneal macrophages.

## Plasmids

The *Tnf* or *Il12b* promoter-driven luciferase reporter plasmids and FLAG-tagged murine NF-κB p65 expression plasmid (pcDNA-FLAG-p65) were designed as described previously [49,50]. The murine MafB cDNA fragment encoding the full-length or the bZip domain deletion mutant was amplified by PCR and subcloned into pCMV-Myc (pCMV-Myc-MafB and pCMV-Myc-MafBdbZip). The FLAG-tagged murine IRF-5 cDNA fragment encoding the full-length or the C-terminal region containing IRF-association domain (IAD) deletion mutant and the HA-tagged murine PU.1 cDNA fragment encoding the full-length or the Ets domain deletion mutant were amplified by PCR and subcloned into pEF-BOS (pEF-BOS-FLAG-IRF5, pEF-BOS-FLAG-IRF5dIAD, pEF-BOS-HA-PU.1, and pEF-BOS-HA-PU.1dEts).

## Luciferase reporter assay

WT or *MAFB*-deficient 293T cells were seeded in 24-well plates ($1.2 \times 10^5$ cells/well), and luciferase reporter and expression plasmids were transiently co-transfected into cells using ScreenFect A. After 24 h, cell lysates were prepared, and luciferase activity was measured using the Dual-Luciferase Reporter Assay System (Promega, Madison, WI, USA).

## Immunoprecipitation and immunoblot analysis

293T cells were transiently transfected with the expression plasmid using Lipofectamine 2000 (Thermo Fisher Scientific, Waltham, MA, USA). After 24 h, collected cells were lysed with a lysis buffer (1% NP-40, 300 mM NaCl, and 20 mM Tris-HCl; pH 7.5) containing a protease inhibitor cocktail (Roche, Basel, Switzerland). The cell lysates were incubated with anti-FLAG M2 mAb (Sigma-Aldrich, St. Louis, MO, USA), anti-Myc-tag mAb (PL14; Medical & Biological Laboratories, Aichi, Japan), or anti-HA-tag pAb (Medical & Biological Laboratories) for 2 h. Tag-specific proteins were subsequently purified using Protein G Mag Sepharose Xtra (GE Healthcare, Chicago, IL, USA). Membranes were incubated with anti-FLAG BioM2-Biotin mAb (Sigma-Aldrich), anti-Myc-tag Biotin mAb (Medical & Biological Laboratories), or anti-HA high-affinity mAb (Sigma-Aldrich) for the immunoblot analysis.

## Reagents

Pam3CSK4 (InvivoGen), all-trans retinoic acid (RA; Sigma-Aldrich), and recombinant IL-10 (Miltenyi Biotec, Bergisch Gladbach, Germany) were used for cell stimulation. MG-132 (Merck, Darmstadt, Germany), bortezomib (Fujifilm Wako Pure Chemical), chloroquine diphosphate salt (Sigma-Aldrich), and anti-IL-10R (Selleck Biotechnology, Kanagawa, Japan) were used as proteasome inhibitors, lysosomal inhibitor, and IL-10 receptor inhibitor, respectively.

## Immunofluorescence analysis

Thioglycolate-elicited peritoneal macrophages were fixed with 4% (wt/vol) paraformaldehyde and permeabilized with 0.5% (vol/vol) Triton X-100. Cells were incubated with rat anti-PU.1 antibody (R&D Systems, Minneapolis, MN, USA), mouse anti-IRF5 antibody (Abcam, Cambridge, U.K.), mouse anti-NFκB p65 antibody (Santa Cruz Biotechnology, Dallas, TX, USA), or rabbit anti-MafB antibody (Abcam). After washing, cells were incubated with Alexa Fluor 488-conjugated anti-rat IgG (Thermo Fisher Scientific), Alexa Fluor 488-conjugated anti-mouse IgG (Thermo Fisher Scientific), and Alexa Fluor 555-conjugated anti-rabbit IgG (Thermo Fisher Scientific) secondary antibodies. Alexa Fluor 594 phalloidin (Thermo Fisher Scientific) and DRAQ5 Fluorescent Probe (Thermo Fisher Scientific) were used to stain actin filaments and the nuclei, respectively. The stained cells were mounted with ProLong Gold antifade reagent (Thermo Fisher Scientific) on glass slides and analyzed using a fluorescence microscope (LSM 700; Carl Zeiss, Oberkochen, Germany).

## Quantitative RT–PCR

Total RNA was isolated using the Sepasol-RNA I Super G reagent (Nacalai Tesque) and reverse-transcribed using the PrimeScript RT reagent kit (Takara Bio, Shiga, Japan). Quantitative RT–PCR was performed using the StepOnePlus Real-Time PCR System (Applied Biosystems, Waltham, MA, USA). The relative gene expression levels were calculated using the $2^{-\Delta\Delta Ct}$ method. All data are shown as the relative mRNA levels normalized to the *Gapdh* level (Table 1).

**Table 1. Primer sequences for quantitative RT–PCR.**

| Gene | Primer sequence (5' to 3') |
| --- | --- |
| *Gapdh*_forward | ACGGCCGCATCTTCTTGTGCA |
| *Gapdh*_reverse | ACGGCCAAATCCGTTCACACC |
| *Mafb*_forward | TGCCTTCTTCTCCCAGCTTC |
| *Mafb*_reverse | AGTGCCTCGGGGGTTCATCT |
| *Rela*_forward | GGCCTCATCCACATGAACTT |
| *Rela*_reverse | CACTGTCACCTGGAAGCAGA |
| *Irf5*_forward | CAGGTGAACAGCTGCCAGTA |
| *Irf5*_reverse | CTCATCCACCCCTTCAGTGT |
| *Spi1*_forward | GGCAGCAAGAAAAAGATTCG |
| *Spi1*_reverse | TTTCTTCACCTCGCCTGTCT |
| *Tnfa*_forward | GGTGATCGGTCCCCAAAGGGATGA |
| *Tnfa*_reverse | TGGTTTGCTACGACGTGGGCT |
| *Il12p40*_forward | CCTGAAGTGTGAAGCACCAA |
| *Il12p40*_reverse | AGTCCCTTTGGTCCAGTGTG |
| *Il1b*_forward | GCCCATCCTCTGTGACTCAT |
| *Il1b*_reverse | AGGCCACAGGTATTTTGTCG |
| *Il6*_forward | TCTGCAAGAGACTTCCATCCAGTTGC |
| *Il6*_reverse | AGCCTCCGACTTGTGAAGTGGT |
| *Nos2*_forward | AACGGAGAACGTTGGATTTG |
| *Nos2*_reverse | TTCTGTGCTGTCCCAGTGAG |
| *Lcn2*_forward | CCAGTTCGCCATGGTATTTT |
| *Lcn2*_reverse | CACACTCACCACCCATTCAG |
| *Slpi*_forward | GTGGAAGGAGGCAAAAATGA |
| *Slpi*_reverse | GACATTGGGAGGGTTAAGCA |
| *Il10*_forward | CCAAGCCTTATCGGAAATGA |
| *Il10*_reverse | TTTTCACAGGGGAGAAATCG |

## ELISA

The concentration of mouse TNF-α (BioLegend, San Diego, CA, USA) and mouse Il-12p40 (Mabtech, Sweden) in culture supernatants was measured by ELISA according to the manufacturer's instructions. The NO detection kit was purchased from iNtRON Biotechnology.

## Bacterial culture and CFU assay

*M. tuberculosis* strain H37Rv was cultured in Middlebrook 7H9 medium (BD, Franklin Lakes, NJ, USA) containing OADC-Enrichment (BD). Control or *Mafb*-deficient peritoneal macrophages were seeded in 6-well plates ($3 \times 10^6$ cells/well) and infected with *M. tuberculosis* (MOI of 5). Harvested cells were plated onto 7H10-OADC agar at the indicated time points after infection.

## *In vivo* experiments

A tracheal tube (Natsume Seisakusho, Tokyo, Japan) was inserted in mouse trachea using a laryngoscope (Welch Allyn, USA), and *M. tuberculosis* solution ($5 \times 10^3$ CFU/ 50 μL) was inoculated via the tracheal tube for the intratracheal challenge. At 0, 1, 2, and 4 weeks after infection, CT and X-ray images were obtained using Cosmo Scan FX in vivo 3D micro-CT system (90 kV; 88 μA; field of view: 25 mm; scan time: 2 min) (Rigaku, Tokyo, Japan) and analyzed using the Analyze software, v.14.0 (AnalyzeDirect, Stilwell, KS, USA). Blood sera collected from mice hearts were used for ELISA, while mice organs homogenized with gentleMACS Dissociator (Miltenyi Biotec) were plated onto 7H10-OADC agar for CFU titers or isolated RNA for quantitative RT-PCR. Ultra-LEAF Purified anti-mouse TNF-α antibody (BioLegend) or Ultra-LEAF Purified anti-mouse IL-12p40 antibody (BioLegend) was inoculated into the abdominal cavity of mice (200 μg/ 250 μL) every other day until 4 weeks of infection for neutralizing experiments.

## Immunohistochemical analysis

Lung tissues were fixed with 20% Formalin Neutral Buffer Solution (Fujifilm Wako Pure Chemical) and sectioned after embedding in paraffin. The sections were then used for hematoxylin and eosin or Ziehl-Neelsen staining. Antigen retrieval was performed by heating sections in 10 mM sodium citrate buffer (pH 6) (for Iba1 and iNOS) or 1 mM EDTA (pH 9) (for Arginase-1) at 95°C for 20 min. After treatment with hydrogen peroxide and blocking with 2% bovine serum albumin in phosphate-buffered saline (PBS) for 30 min, the sections were incubated with primary antibodies at 4°C overnight. The primary antibodies used were rabbit antibodies for Iba1 (1:500, Fujifilm Wako Pure Chemical), Arginase-1 (1:100, Protein-Tech, Chicago, IL, USA), and iNOS (1:100, Novus Biologicals, Centennial, CO, USA). After washing the sections with PBS, staining was achieved with a Simple Stain kit (Nichirei, Tokyo, Japan) and developed with 3,3'-diaminobenzidine tetrahydrochloride (DAB) and hydrogen peroxide (Nichirei) at RT for 5–7 min. The sections were counterstained with hematoxylin.

## Statistical analysis

Statistical analyses were performed using the Student's t-test, one-way ANOVA, or Log-rank test. Values of $P < 0.05$ were considered statistically significant.

## Supporting information

**S1 Fig. Expression of relative transcription factors in *Mafb*-deficient peritoneal macrophages.** Mice were intraperitoneally injected with 4% thioglycolate, and macrophages were isolated from the peritoneal cavity three days later. Cells were harvested and analyzed using quantitative RT–PCR. Relative gene expression is represented as a fold increase above background level of samples prepared from control cells. Data are representative of two independent experiments. ***$P < 0.001$; n.s., not significant; mean ± SD.
(TIF)

**S2 Fig. mRNA expression and cytokine production in Mafb-deficient peritoneal macrophages. (A)** Mice were intraperitoneally injected with 4% thioglycolate, and macrophages were isolated from the peritoneal cavity three days later. Following a 6-h stimulation with 100 ng/mL Pam3CSK4, cells were harvested and analyzed using quantitative RT–PCR. Relative gene expression is represented as a fold increase above background level of samples prepared from unstimulated cells. Data are representative of three independent experiments. **$P<0.01$; ***$P<0.001$; mean ± SD. **(B)** Mice were intraperitoneally injected with 4% thioglycolate. Three days later, macrophages were isolated from the peritoneal cavity. Following a 6-h stimulation with 0, 100, 200, 400, or 500 ng/mL Pam3CSK4, cytokine production was measured using ELISA. Data are representative of three independent experiments. *$P<0.05$; **$P<0.01$; ***$P<0.001$; n.s., not significant; mean ± SD. (TIF)

**S3 Fig. Expression of bactericidal inflammatory effectors in *Mafb*-deficient peritoneal macrophages.** Mice were intraperitoneally injected with 4% thioglycolate, and macrophages were isolated from the peritoneal cavity three days later. At the indicated time points after infection with *M. tuberculosis* (MOI of 5), cells were harvested and analyzed using quantitative RT–PCR. Relative gene expression is represented as a fold increase above background level of samples prepared from uninfected control cells. Data are representative of three independent experiments. (TIF)

**S4 Fig. Expression of MafB mRNA upon inhibiting IL-10 signaling or nuclear receptors.** Mice were intraperitoneally injected with 4% thioglycolate, and macrophages were isolated from the peritoneal cavity three days later. Cells were infected with *M. tuberculosis* H37Rv (MOI of 5) after priming with anti-mouse IL-10R (0, 10, 20, 50, or 100 μg/mL), AR7 (RAR inhibitor; 0, 10, 20, 50, or 100 μM), or GW9662 (PPAR-γ inhibitor; 0, 10, 20, 50, or 100 μM). Six hours following infection, cells were harvested and analyzed using quantitative RT–PCR. *Mafb* mRNA level is normalized by the corresponding *Gapdh* level. Data are representative of two independent experiments. ***$P<0.001$; mean ± SD. (TIF)

**S5 Fig. Histopathological analysis of the lungs in mycobacteria-infected mice.** *Mafb*^f/f^ (n = 3) and *Mafb*^f/f^::LysM-Cre (n = 3) mice were intratracheally infected with *M. tuberculosis* H37Rv ($5 \times 10^3$ CFU). Three weeks after infection, the lungs were fixed in paraffin, and lung tissue sections were used for several staining procedures. **(A)** The number of nodular lesions was determined using hematoxylin-eosin-stained lung sections. *$P<0.05$. **(B)** The degree of nodularity was scored using Iba1 immunohistochemical staining panels. 3: almost nodal, 2: nodal predominance, 1: diffuse predominance. *$P<0.05$. **(C)** The degree of TB area was scored using Ziehl-Neelsen staining panels. 3: positive in areas > 50%, 2: positive in the 25%–50% range, 1: positive in areas < 25%. n.s., not significant. **(D)** The number of Arginase-1-positive macrophages was determined using Arginase-1 immunohistochemical staining panels (Scale bar represents 100 μm). ***$P<0.001$. **(E)** The degree of iNOS-positive macrophages area was scored using iNOS immunohistochemical staining panels (Scale bar represents 100 μm). 2: positive in areas > 50%, 1: positive in areas < 50%. *$P<0.05$. (TIF)

**S1 Data. Raw data.** This file includes raw data of Figs 1A, 1B, 1C, 1D, 2D, 2G, 2H, 3B, 3C, 3D, 3E, 3F, 3G, 4A, 4B, 4C, 4D, 4E, 4F, 4G, 4H, 4I, 4J, 4K, 4L, 4M, 4N, 5A, 5B, 5C, 5D, 5G, 5H, S1, S2A, S2B, S3, and S4. (XLSX)

**S2 Data. Raw image.** This file includes raw image data of Figs 2A, 2B, 2C, 2E, 2F, 3A, 3C, 3E, 4B, 4E, 5E, 5F, S5A, S5B, S5C, S5D, and S5E. (PDF)

## Acknowledgments

We thank Makiko Suzuki for technical assistance with the experiments and Eriko Tanaka, Manami Uematsu, Ryoko Itami, and Junko Kitamura for secretarial assistance. We also thank Machi Kawauchi for technical assistance with immunohisto-chemical analysis, Mizuho Kujiraoka for animal care and breeding, and Editage (www.editage.jp) for editing a draft of this manuscript. Finally, we express our special thanks to Stefan H. E. Kaufmann (Max Planck Institute for Infection Biology, Germany) for his valuable comments and suggestions on our manuscript.

## Author contributions

**Conceptualization:** Hiroyuki Saiga, Katsuaki Hoshino, Manabu Ato.

**Data curation:** Hiroyuki Saiga.

**Formal analysis:** Hiroyuki Saiga, Masaki Ueno.

**Funding acquisition:** Hiroyuki Saiga, Yusuke Tsujimura, Manabu Ato.

**Investigation:** Hiroyuki Saiga, Masaki Ueno, Toshiki Tamura, Yusuke Tsujimura, Masamitsu N Asaka, Yumiko Tsukamoto.

**Project administration:** Hiroyuki Saiga.

**Resources:** Toshiki Tamura, Tetsu Mukai, Michito Hamada, Satoru Takahashi, Takashi Tanaka, Tsuneyasu Kaisho, Katsuaki Hoshino.

**Supervision:** Yoshimasa Takahashi, Katsuaki Hoshino, Manabu Ato.

**Validation:** Hiroyuki Saiga, Masaki Ueno.

**Visualization:** Hiroyuki Saiga, Masaki Ueno.

**Writing – original draft:** Hiroyuki Saiga, Masaki Ueno.

**Writing – review & editing:** Hiroyuki Saiga, Masaki Ueno, Toshiki Tamura, Yusuke Tsujimura, Masamitsu N Asaka, Yumiko Tsukamoto, Tetsu Mukai, Michito Hamada, Satoru Takahashi, Takashi Tanaka, Tsuneyasu Kaisho, Yoshimasa Takahashi, Katsuaki Hoshino, Manabu Ato.

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
