## [Decision Letter · Decision Letter 0]

7 Feb 2025

PPATHOGENS-D-25-00053

*Mycobacterium tuberculosis* impairs cytokine production by manipulating transcription factor MafB

PLOS Pathogens

Dear Dr. Saiga,

Thank you for submitting your manuscript to PLOS Pathogens. After careful consideration, we feel that it has merit but does not fully meet PLOS Pathogens's publication criteria as it currently stands. Therefore, we invite you to submit a revised version of the manuscript that addresses the points raised during the review process. As you can see in the attached reviews, the referees appreciated the importance of the topic and the novelty of your observations.  However, the reviewers agreed that the study was not yet conclusive, and still contained discrepancies that need to be addressed.  Several suggestions were offered to further characterize either the signals regulating MafB, or its downstream effects.  While we do not expect every suggested experiment to be performed, it is clear that significant additional work on these topics is warranted. 

Please submit your revised manuscript within 60 days Apr 08 2025 11:59PM. If you will need more time than this to complete your revisions, please reply to this message or contact the journal office at plospathogens@plos.org. Please include the following items when submitting your revised manuscript:

We look forward to receiving your revised manuscript.

Kind regards,

Christopher M. Sassetti

Academic Editor

PLOS Pathogens

Michael Wessels

Section Editor

PLOS Pathogens

 Sumita Bhaduri-McIntosh

Editor-in-Chief

PLOS Pathogens

orcid.org/0000-0003-2946-9497

 Michael Malim

Editor-in-Chief

PLOS Pathogens

orcid.org/0000-0002-7699-2064

**Journal Requirements:**

At this stage, the following Authors/Authors require contributions: Hiroyuki Saiga, Masaki Ueno, Toshiki Tamura, Yusuke Tsujimura, Masamitsu N Asaka, Yumiko Tsukamoto, Tetsu Mukai, Michito Hamada, Satoru Takahashi, Takashi Tanaka, Tsuneyasu Kaisho, Yoshimasa Takahashi, Katsuaki Hoshino, and Manabu Ato. Please ensure that the full contributions of each author are acknowledged in the "Add/Edit/Remove Authors" section of our submission form.

3) Some material included in your submission may be copyrighted. According to PLOSu2019s copyright policy, authors who use figures or other material (e.g., graphics, clipart, maps) from another author or copyright holder must demonstrate or obtain permission to publish this material under the Creative Commons Attribution 4.0 International (CC BY 4.0) License used by PLOS journals. Please closely review the details of PLOSu2019s copyright requirements here: PLOS Licenses and Copyright. If you need to request permissions from a copyright holder, you may use PLOS's Copyright Content Permission form.

Potential Copyright Issues:

- Please confirm that you are the photographer of Figure 5E, or provide written permission from the photographer to publish the photo under our CC BY 4.0 license.

4) Please ensure that the funders and grant numbers match between the Financial Disclosure field and the Funding Information tab in your submission form. Note that the funders must be provided in the same order in both places as well. State what role the funders took in the study. If the funders had no role in your study, please state: "The funders had no role in study design, data collection and analysis, decision to publish, or preparation of the manuscript.".

**Reviewers' Comments:**

Reviewer's Responses to Questions

**Part I - Summary**

Reviewer #1: This paper by Saiga et al works to understand the role of MAFB in M. tuberculosis pathogenesis, as it has been associated with TB susceptibility in genetic studies of Mtb growht. They focus on the role of MAFB in the macrophage in this paper, demonstrating in elegant studies that it inhibits transcription of proinflammatory cytokines by preventing binding of p65 to initiate transcription. Further, they demonstrate that deletion of MAFB selectively in macrophages improves M.tb outcomes and increases the proportion of proinflammatory macrophages in the lung. These observations are interesting and potentially important for understanding m.tb pathogenesis. However, the discussion is sparse and the characterization of MAFB-/- macrophages is incomplete, espeically the role of MAFB in Type I intereforon secretion, as they showed in prior studies that this regulates Type I interferon production in plasmacytoid DC.

Reviewer #2: Saiga et al examine how the transcription factor MafB impacts macrophage cytokine responses and the host response to Mycobacterium tuberculosis. The authors determine that MafB directly interacts with transcription factors IRF5 and PU.1 to inhibit pro-inflammatory cytokines TNF and IL-12p40 and demonstrate, using a conditional knockout mouse model, that the lack of MafB expression in macrophages leads to decreased bacterial burden and increased survival following Mtb infection. The dissection of MafB molecular interactions is particularly strong. The authors generate and use domain mutants for MafB, IRF5, and PU.1 and express these in HEK293T cells to determine direct interactions between transcription factors. They also use both an in vitro model of elicited peritoneal macrophages and an in vivo mouse model using MafB conditional knockout mice to test the role of MafB in macrophages during Mtb infection.

Overall, identifying the factors that regulate whether Mtb-infected macrophages mount a pro-inflammatory response or not is critical to understanding host-pathogen interactions during Mtb infection. Therefore, the overall topic is of significance to the field. This manuscript provides new information about a source of macrophage regulation that was not previously described during Mtb infection. Overall, the studies are well-conceived, with adequate controls. The manuscript is well-written and the experiments are clearly described. However, there are several current limitations to the manuscript that limit enthusiasm, including the rigor of the statistical analysis and some discrepancies between experiments, that need to be addressed.

Reviewer #3: This study investigates the role of the transcription factor MafB in TB pathogenesis, particularly its impact on cytokine production. The authors demonstrate that Mtb upregulates MafB to suppress pro-inflammatory cytokines, such as TNF-α and IL-12p40, thereby facilitating bacterial survival. Using MafB-deficient mice, the study shows enhanced cytokine production, reduced bacterial burden, and improved host resistance, suggesting MafB as a potential biomarker for TB susceptibility. However, the mechanistic basis for MafB regulation remains incompletely defined, particularly regarding the upstream signaling pathways controlling its expression during Mtb infection. Addressing these mechanistic gaps would further solidify the study’s conclusions and translational relevance.

**Part II – Major Issues: Key Experiments Required for Acceptance**

Reviewer #1: Line Comment

213 Additional data in macrophages with MAFB overexpression would be good to validate ATRA experiments. It may be that RA induces cytokine responses via an alternate non-MAFB mechanism. Or an experiment that demonstrates that MAFB knockout abolishes ATRA proinflammatory effects

330 What is the breadth of the response in the absence of MAFB? Does MAFB influence other TLR activation? If TLR2 deficient mice do not display worsened m.tb diseaes, why then does MAFb alter responses?

Reviewer #2: 1) The statistical analysis in the manuscript lacks rigor and critical details. Many bar and line graphs lack error bars. The numbers of technical or biological replicates should be reported for each subfigure. “ * ” statistics shown in all figures do not indicate which comparisons are significant. This information needs to either be provided within the figure via black bars between specific conditions or clearly described within the figure legend or text.

2) The manuscript would benefit from a more robust investigation as to why MafB expression is induced during Mtb infection. The authors wait for the discussion before offering a hypothesis for why MafB expression goes up during Mtb infection, stating that expression of IL-10 might be responsible for induction of MafB. In the discussion, Fig S4 is shared to support this claim. This information should be provided in the main text as part of Fig 4 and it would be helpful to have more data provided to support this claim. For example, does anti-IL-10R treatment block the Mtb-induced MafB expression increase? Why does IL-10-mediated induction of MafB not play a role during TLR stimulation, in which the authors observe that MafB expression decreases, as TLR stimulation can also induce IL-10? The authors provide an alternative hypothesis that nuclear receptors, specifically retinoic acid receptor or PPARg, could induce MafB expression. The authors could similarly test these mechanisms.

3) Line 326: “the MafB expression was not detected in MafB cKO lungs (Fig 5A), indicating that MafB expression in lungs is dependent on tissue macrophages.” The MafB expression data for the cKO lungs was not provided and should be included in Figure 5, if the authors want to make the above claim.

Issues to address in the text:

1) In figure S2A, cKO macrophages have greater Il1b and Il6 expression following TLR2 agonist stimulation but decreased Il1b and Il6 expression following Mtb infection in Fig 4G. The authors should address this discrepancy.

2) While the molecular dissection of MafB binding in Figures 1 and 2 is very strong, the luciferase reporter assay and immunoprecipitation were performed using HEK293T cells, which may not be the best representative for macrophage intracellular interactions. At minimum, the authors should include a discussion of how the TF interactions might differ between cell types or cell activation states.

Reviewer #3: Here are some of my primary concerns, which would require additional experiments.

1. The study extensively relies on 293T cells to examine MafB’s interactions and transcriptional regulation. While these assays provide mechanistic insights, they lack physiological relevance, as 293T cells do not fully recapitulate macrophage-specific immune responses. Key experiments should be replicated in primary macrophages under Mtb infection conditions to confirm the functional role of MafB in a more relevant context.

2. The manuscript does not clarify whether MafB regulation is primarily mediated through TLR2 or TLR4 signaling. Given that TLR4 (activated by LPS) and TLR2 (recognized by Mtb components) initiate distinct immune responses, it is crucial to determine which pathway predominantly controls MafB expression during Mtb infection. Experiments using TLR2- or TLR4-deficient macrophages would be necessary to provide definitive insights.

3. The study reports that TLR2 stimulation with Mtb-derived components decreases MafB levels while increasing TNF-α and IL-12p40 production. However, during live Mtb infection, MafB expression instead increases. Since Mtb contains TLR2 agonists, this apparent contradiction needs further clarification. Does Mtb actively counteract TLR2-induced MafB suppression through additional virulence mechanisms? Investigating whether Mtb employs alternative pathways to upregulate MafB would help reconcile these findings.

4. The manuscript states that TNF-α is undetectable in in vivo experiments, yet neutralizing TNF-α has a significant impact on bacterial burden. This raises concerns about the detection method used—was the sensitivity of the assay sufficient? Alternatively, could other factors, such as localized TNF-α production within granulomas, explain this discrepancy?

5. The study does not address whether TNF-α or IL-12p40 plays a more dominant role in MafB-mediated immune suppression. Since both cytokines are critical for TB immunity, it would be valuable to assess the effects of individual vs. combined neutralization of TNF-α and IL-12p40 in MafB-deficient mice. This could help determine whether one cytokine is the primary driver of resistance in the absence of MafB. Additionally, does TNF-α or IL-12p40 neutralization in wild-type mice fail to enhance Mtb pathogenicity?

**Part III – Minor Issues: Editorial and Data Presentation Modifications**

Reviewer #1: The discussion is cursory and does not provide context for several issues of interest -- why does MAFB improve MTB outcomes when other adaptors that increase immune responses, like TOLLIP (Venkatasubramanian et al Nat Micro 2024) or variants in IL1B (Zhang et al Plos Pathogens 2014), do not? Does the mouse model recapitulate findings in human populations -- have MAFB been shown to have a functional variant and does the directionality of effect between mice and humans correlate? If not, why not?

Recently, observation were made that excess Type I interferon was detrimental for Mtb control (Ji et al Nat Micro 2019). Given that MAFB deficiency in macropahges is beneficial with increased TNF

Does MAFB induce increased Type I inteferon in macrophages after Mtb infection as it does with pDC? This may be key to understanding why increased cytokines in this model are beneficial while they are detrimental in other models

Reviewer #2: 1) Line 92: change “molecular basis evidence” to “molecular-based evidence”

2) The macrophages used in Figures 3 and 4 should be correctly described as “thioglycolate-elicited peritoneal macrophages” or “elicited peritoneal macrophages”.

3) The different conditions in 5G are difficult to see, especially the “cKO + anti-TNFa” condition, which is hidden behind the other conditions.

4) Pathology and lesions in 5E would benefit from quantification, if possible.

5) Figure 6 should depict the role of MafB during Mtb infection (in addition to TLR stimulation).

Reviewer #3: none.

PLOS authors have the option to publish the peer review history of their article (what does this mean? ). If published, this will include your full peer review and any attached files.

**Do you want your identity to be public for this peer review?** For information about this choice, including consent withdrawal, please see our Privacy Policy .

Reviewer #1: No

Reviewer #2: No

Reviewer #3: No

**Figure resubmission:**
---

## [Decision Letter · Decision Letter 1]

21 Aug 2025

Dear Dr. Saiga,

We are pleased to inform you that your manuscript '*Mycobacterium tuberculosis* impairs protective cytokine production via transcription factor MafB manipulation' has been provisionally accepted for publication in PLOS Pathogens.

Best regards,

Christopher M. Sassetti

Academic Editor

PLOS Pathogens

Michael Wessels

Section Editor

PLOS Pathogens

Sumita Bhaduri-McIntosh

Editor-in-Chief

PLOS Pathogens

orcid.org/0000-0003-2946-9497

Michael Malim

Editor-in-Chief

PLOS Pathogens

orcid.org/0000-0002-7699-2064

Reviewer Comments (if any, and for reference):

Reviewer's Responses to Questions

**Part I - Summary**

Reviewer #1: Thank you for your thoughtful and thorough response to the reviews.

Reviewer #2: The resubmitted manuscript "Mycobacterium tuberculosis impairs protective cytokine production via transcription factor MafB manipulation” by Saiga et al has been significantly improved over the original submission. All major and minor concerns have been addressed.

Reviewer #3: (No Response)

**Part II – Major Issues: Key Experiments Required for Acceptance**

Reviewer #1: (No Response)

Reviewer #2: (No Response)

Reviewer #3: (No Response)

**Part III – Minor Issues: Editorial and Data Presentation Modifications**

Reviewer #1: (No Response)

Reviewer #2: (No Response)

Reviewer #3: (No Response)

PLOS authors have the option to publish the peer review history of their article (what does this mean? ). If published, this will include your full peer review and any attached files.

**Do you want your identity to be public for this peer review?** For information about this choice, including consent withdrawal, please see our Privacy Policy .

Reviewer #1: No

Reviewer #2: No

Reviewer #3: No

---

## [Editor Report · Acceptance letter]

Dear Dr. Saiga,

We are delighted to inform you that your manuscript, "*Mycobacterium tuberculosis* impairs protective cytokine production via transcription factor MafB manipulation," has been formally accepted for publication in PLOS Pathogens.

Best regards,

Sumita Bhaduri-McIntosh

Editor-in-Chief

PLOS Pathogens

orcid.org/0000-0003-2946-9497

Michael Malim

Editor-in-Chief

PLOS Pathogens

orcid.org/0000-0002-7699-2064